# Wearables, Biomechanical Feedback, and Human Motor-Skills' Learning & Optimization

**Xiang Zhang [1,†], Gongbing Shan [1,2,3,*,†], Ye Wang [4], Bingjun Wan [3] and Hua Li [4]**

1 Department of Physical Education, Xinzhou Teachers' University, Xinzhou 034000, China; xiangzhang@xztc.edu.cn
2 Biomechanics Lab, Faculty of Arts & Science, University of Lethbridge, Lethbridge, AB T1K 3M4, Canada
3 School of Physical Education, Shaanxi Normal University, Xi'an 710119, China; bingjunw55@snnu.edu.cn
4 Department of Mathematics & Computer Science, University of Lethbridge, Lethbridge, AB T1K 3M4, Canada; ye.wang3@uleth.ca (Y.W.); hua.li@uleth.ca (H.L.)
* Correspondence: g.shan@uleth.ca; Tel.: +1-403-329-2683
† The authors contributed equally to this work.

**Abstract:** Biomechanical feedback is a relevant key to improving sports and arts performance. Yet, the bibliometric keyword analysis on Web of Science publications reveals that, when comparing to other biofeedback applications, the real-time biomechanical feedback application lags far behind in sports and arts practice. While real-time physiological and biochemical biofeedback have seen routine applications, the use of real-time biomechanical feedback in motor learning and training is still rare. On that account, the paper aims to extract the specific research areas, such as three-dimensional (3D) motion capture, anthropometry, biomechanical modeling, sensing technology, and artificial intelligent (AI)/deep learning, which could contribute to the development of the real-time biomechanical feedback system. The review summarizes the past and current state of biomechanical feedback studies in sports and arts performance; and, by integrating the results of the studies with the contemporary wearable technology, proposes a two-chain body model monitoring using six IMUs (inertial measurement unit) with deep learning technology. The framework can serve as a basis for a breakthrough in the development. The review indicates that the vital step in the development is to establish a massive data, which could be obtained by using the synchronized measurement of 3D motion capture and IMUs, and that should cover diverse sports and arts skills. As such, wearables powered by deep learning models trained by the massive and diverse datasets can supply a feasible, reliable, and practical biomechanical feedback for athletic and artistic training.

**Keywords:** anthropometry; biomechanical modeling; two-chain body model; joints' coordination; IMUs; deep learning

---

Wearable sensors have garnered great interest in biofeedback training, owing to their tremendous promise for a plethora of applications. They supply real-time non-invasive monitoring of physical-activity parameters as indicators of a trainee's physical progress. Yet, the absence of a reliable method of applying wearables in biomechanical feedback training has greatly hindered wearable application in the area of human motor skill learning, training, and optimization. This review article intends to concentrate on the theme of biomechanical feedback training, selecting the relevant/related articles to summarize previous investigations in order to inform the reader of the current state and the next possible steps for its development. The article reports on:

- Biofeedback and its types
- Biomechanical feedback in motor learning—how far is it from a real-time application?
- Milestones of biofeedback training in human motor-skill learning

- Defines and clarifies the problem—why is biomechanical one different from other biofeedback?
- Significant gaps in the current research
- How past and current developments influencing new endeavors—ideas of where research might break through.

## 1. Biofeedback and Its Types

Biofeedback is usually gained by connecting human body to electrical sensors that receive information (feedback) about human body (bio). It is a technique that one can use to learn to control one's body functions or physical performance [1]. Generally, there are four types of biofeedback: physiological (e.g., heart rate and blood pressure), neurological (e.g., electroencephalogram (EEG)/brain-wave), biochemical (e.g., electrolytes and metabolites in sweat or saliva), and biomechanical (e.g., joint angles and force applied) [2,3]. In human motor learning, biofeedback training familiarizes us with the activity in our various body systems, so it is an useful educational and/or training tool for mastering and/or maintaining human motor skills [4].

## 2. Biomechanical Feedback in Motor Learning—How Far Is It from a Real-Time Application?

Wearables in sports are only few years old; however, they have expanded radically, from the real-time monitoring of players' signs of exhaustion or injury while on the field to including perceptual and psychological aspects of professional team sports for enhancing performance [5–7]. It has already been a public's agreement that wearable technology is leading a revolution in sports [5,7,8]. Various sensors are now fitted into sport equipment, wristband, and/or clothing to determine athletic performance, like speed, acceleration, power, distance, heart, and metabolic conditions during training. All the crucial data are sent to the coach and training team instantly, allowing for them to perform an individualized training for increasing athletic competence.

Nevertheless, a real-time biomechanical feedback training would currently not look so optimistic. A search using keywords in the authority database—Web of Science—reveals the following scenario: When the keyword "biofeedback training" is applied, 5588 articles are found. However, when the keyword is changed to "biomechanical feedback training", the article number is dramatically dropped to 569. Even more theatrical, when two additional keywords "real-time" and "sport" are added for a search, the number decreased to 23. To the end of the search, scarcity of articles occurs when the keyword "sport" is substituted by "dancing", i.e., only one article is found (Table 1). These results would suggest that, when comparing to other biofeedback applications, the real-time biomechanical feedback application lag far behind in sports and arts practice.

**Table 1.** The result of literature search in all databases of Web of Science on 11 October 2018.

| Biofeedback Training | Biomechanical Feedback Training | Biomechanical Feedback Training & Real-Time & Sport | Biomechanical Feedback Training & Real-Time & Dancing |
|:---:|:---:|:---:|:---:|
| 5588 | 569 | 23 | 1 |

A close look at the types of published papers divulges that the real-time biomechanical feedback training in motor learning is still an infant science, i.e., only two applied studies attempting to reveal its potentials in human motor learning/training (Table 2). When considering the booming popularity of wearables in sports as well as in health-related applications, the number of biomechanical inquiries appears to be disproportionately low. The rarity of this occurrence could be a product of both facts that there is a lack of a general biomechanical model for feedback motor learning and that researchers are still searching for methodological breakthroughs to link biomechanical quantification and human motor learning in real-time.

**Table 2.** The article types of real-time biomechanical feedback training found in Web of Science.

| | Motor Learning/Training | Method/Development | Injury Prevention/Rehabilitation | Review Articles | Patents | Total |
|---|---|---|---|---|---|---|
| Sport | 2 | 10 | 7 | 2 | 2 | 23 |
| Dancing | 0 | 1 | 0 | 0 | 0 | 1 |

## 3. Milestones of Biofeedback Training in Human Motor Skill Learning and Training

Effective human motor skill learning/training benefits nearly every one of us, as it can help develop interests in more physical activities and lead to more active lifestyles [9]. The main aims of researches that are related to human motor skills' learning (both in sports and arts performance) are to improve learning techniques (education), to accelerate skill acquisition (learning), and to maintain motor function (training). All the three aspects rely on the feedback mechanism for their efficiency and effectiveness [10]. Given the complexity of human sensory-motor behavior, informed learning and training hold a great potential to improve efficiencies, particularly in the acquisition of cognitive and psychomotor skills for highly complicated performance activities [11–15]. The two key components in human motor skill learning and training are practice and biofeedback [16]. Previous studies have shown that, when properly understood and applied, biofeedback is an excellent tool for enhancing practice and performance of human motor skills [17–23].

### 3.1. Historical Overview

Learning and training of human motor skills has a history of over thousands of years [15,16], experiencing some key periods, such as apprenticeship, class education, individualized instruction, and integrated performance support; however, the appearance of (bio)feedback in systematic motor skill training occurred in early 1950s [24,25]. After World War II, individualized instruction was first developed in industry for training human physical skills (i.e., human motor skills) efficiently and reducing expense while still getting high instructional value for various professionals. The training method broke the learning into small steps with an activity afterward to check comprehension. The reinforcement learning behavior opened the door for biofeedback intervention in motor learning and practicing new motor skills.

This early form of feedback learning in essence requires immediate feedback (i.e., real-time feedback) given after each skill practice. The training can be knowledge-based (trainer), or more objectively, technology-based. The advantages of the feedback learning are: (1) it allows for a learner to practice at his or her own pace and to find mistakes and correct them and (2) it reduces learning time, produces a low error rate, and improves learning efficiency through immediate feedback [15,16]. The successful example of the feedback learning is a computer-based training developed and is used primarily in the military [25]. The benefits of such training are more opportunities for realistic training and feedbacks; and, increased availability and accessibility of training in operational units.

From scientific point of view, human motor-skill development is a biological process; therefore, the influential feedbacks should be those related to the changes of biological parameters of human motor system. In essence, feedback in human motor-skills' training is primarily biofeedback. Biofeedback as a research major was first reported in the 1960s, supplying single-parameter feedback in real-time training [2]. Until the end of the last century, biofeedback had been able to supply multiple parameters, such as body temperature, heart rate, respiratory rate, muscle activity, impact, joint angle, and others during training [26–28]. Only due to the limitation of sensing technology at that time, the application was commonly lab-based and applicants were equipped with wires. As such, the applications were mainly in areas of less human mobility or less human movement complexity, such as in senior health care, physiotherapy, and rehabilitation [2,29–33], i.e., the application was lab-based, not practical in motor learning/training related to sports and arts performance.

*3.2. The Present Aspects*

Over the past decade, wearables are becoming the trend in sports training. The technological developments have led to the production of inexpensive, non-invasive, miniature sensors, which are ideal for obtaining sport performance measures during training or competition. The miniature sensing devices are worn on wrist, clothes, and/or shoes. They supply real-time biofeedback for sports analyses. The sensing technology has turned towards creating devices with new form factors that augment sports activities.

The overwhelmingly impression of wearables success in sports is mainly in motoring physical condition and preventing injuries. For sport-related injuries, soft-tissue injury remains the most common type among athletes. The injury is often caused by fatigue, overtraining, or dehydration [34–36]. Wearable sensors are now able to collect data related to these risk physical conditions from athletes' physical conditions, muscle activities, and sweat [3,5–7]. The real-time biofeedback helps coaches to alternate quickly their training or competition strategies for decreasing this major injury in trainings and competitions [5,7,34].

*3.3. The Current Success of Wearables in Sports Is Not Yet Linked to the Human Motor-Skill Learning*

The existing evidences demonstrate that the wearables have successfully supplied real-time information related to athletes' speed, acceleration, power, distance (i.e., locomotion/physical characteristics), heart rate, muscle activities (i.e., physiological feedback), and electrolytes, metabolites (i.e., biochemical feedback). Although, these parameters are useful in analyzing the general physical condition of an athlete, they do not provide information that is related to the limbs' control of human motor skills, i.e., the biomechanical feedback is still missing. Without this vital information, the motor learning of complicate skills (e.g., artistic performance, gymnastics/acrobatics skills, and many others) is largely formed of art based on the trainers' subjective experiences of "what works" [11,12,37]. While this can be effective for some learners, large and widespread biological diversity unfortunately limits the generalizability of a single individual's experiences [11,12,38,39]. Even small variations in bone length, muscle, and tendon attachments, for example, can disrupt this traditional form of knowledge transfer. Thus, to improve motor-skills' learning, we need to establish scientifically described training targets and routes, which in turn require biomechanical feedback tools that can measure and quantify characteristics of effective limbs' coordination (i.e., motor control).

## 4. Defines and Clarifies the Problems—Why Is Biomechanical One Different from Other Biofeedback?

*4.1. Unique Aspects of Biomechanical Feedback*

Physiological, neurological, and biochemical feedback present information related to one's physiological variation, muscle tension, physical condition, and thought processes. Such information is conserved across human motor skills, i.e., across different movement forms. Therefore, feedback devices monitoring these parameters can be universally applied to all activities [40,41].

In contrast to physiological, neuronal, and biochemical feedback, biomechanical feedback mainly provides information that is related to the limb control of human motor skills, which directly accelerates motor skill learning and optimization, but must be tailored to the activity being examined [42–45]. In other words, biomechanical feedback is thus a more useful tool but complicated for its development.

Several studies in the past decades confirmed the importance of real-time biomechanical feedback, showing up to 100% improvement with its application [46–49]. However, the development of biomechanical feedback is still in its infancy. While the real-time biofeedback of the first three types (i.e., physiological, neurological, and biochemical feedback) has been well developed for the past decades and is now a routine application (successfully transferred from lab-based to training and/or competition environments), the studies and applications of biomechanical one are still rare. After reviewing 666 publications between 1960's and 2010's, Tate and his colleagues found that there

were only seven studies using real-time biomechanical feedback for physical training in laboratory environment [2]. Supplementary, the current state has not shown a considerable change, especially in sports and arts performances (Tables 1 and 2). The rarity could be caused by the numerous hitches that must be overcome during the development of the real-time biomechanical feedback tools. The obvious one is that biomechanical feedback must always be tailored to an activity (i.e., non-generalizable), requiring different design parameters for different motor skills. Thus, to develop a biomechanical feedback device, one must first obtain a thorough understanding of the selected motor skill in order to select the useful parameters for monitoring. Additionally, the devices must not interfere with the motor skill being executed. This technical limitation alone has proved to be a major hindrance to the development of biomechanical feedback devices in motor learning and training.

In short, biomechanical wearables still require much more researches before it can become an impactful tool in the real world.

### 4.2. Biomechanical Steps Required in Developing Wearables for Biomechanical Feedback

As discussed before, a successful motor learning outcome can be supported by useful and timely biomechanical feedback to the athlete targeting performance defects. Systematic, objective, and reliable performance monitoring and evaluation, performed by means of quantitative analyses of biomechanical variables, can reinforce the biomechanical feedback training in sports practice [12,37,42]. Therefore, the ways of quantifying a motor skill with high spatial and temporal accuracy (i.e., the limbs coordination) would be the key for developing wearables for biomechanical feedback training [50].

Currently, the most reliable biomechanical feedback method is three-dimensional (3D) motion capture, which identifies and tracks markers that are attached to a human subject's joints and body parts to obtain 3D skeleton information [51–53]. The spatiotemporal human representation based on 3D motion capture data is currently the most trustworthy approach in motor skill quantification, both in sports and arts performance [54–60]. This method, however, mainly supplies post-measurement feedback (i.e., not real-time) due to its drawbacks: multiple cameras placed in a room, long calibration and setup procedures, a time consuming course on data collection, processing, analysis, and interpretation, and the high cost of the equipment [61–63].

For practitioners, real-time feedback is more useful. Yet, due to the drawbacks of 3D motion analysis technology and the diversity of human motor skills in sports and arts performance, research on biomechanical feedback has to undergo:

(1)  selection of a specific motor skill,
(2)  3D motion analysis of the skill,
(3)  verification of post-measurement feedback in practice, and
(4)  development of feedback device for monitoring the critical/vital parameter(s) (e.g., coordination among certain segments or joints) for the given motor skill.

These steps are, at present, required for developing a reliable device that is capable of supplying real-time biomechanical feedback [27,50].

### 4.3. Challenges Faced by Developing Wearables for Biomechanical Feedback

The current sensing development has shown its potential to mitigate problematic constraints of biofeedback devices on human movement and demonstrated its great promise to expand the capabilities of biofeedback to motor-skill learning [5]. The successes in health and physiotherapy [26,29,64,65] suggest that biomechanical wearables will become a reality in human motor learning and training in sports and arts. However, the transition from the simple motor skills training to the complicated ones would face several challenges.

It is no doubt that the most challenge for developing biomechanical wearables is the practicality in sports and arts performance. Any device attached to human body will supply certain constraints for our movement and alternate the movement control in a way that may not reach the goal of

training. Currently, the reliable 3D motion capture technology requires around 40 markers for motor skills quantification and characterization [52,59,60,66]. Even the non-ideal test condition cannot be substituted by simply replacing the ~40 markers (sphere shape of 9 mm in diameter and almost weightless) with wearables, because the weight and volume of current wearables (e.g., IMUs) can cause unknown experimental artifacts. Therefore, how to apply fewer wearables (e.g., 4–6) for accurately rebuilding sports and arts motor skills would be the primary focus for the development of wearables in human motor-skill learning. While the use of the ~40 motion-capture markers can directly quantify complicated motor skills, there is no direct way for using 4-6 wearables to reach the same goal.

The second challenge is the identification of motor control patterns. In sports and arts activities, the motor control patterns exhibit the characteristic wherein either gross or fine motor control appears to be dominant. In most sports, it is reasonable to conceive that the majority of activities (e.g., running, jumping and throwing) mainly rely upon large muscles (i.e., gross motor control), where smaller muscles function in significant stabilizing roles. Fewer activities, like shooting, rely mainly on smaller muscle group coordination (i.e., fine motor control) where gross motor control supplies foundational support or is nearly arrested [44,51]. In the case of music performance (e.g., playing the piano or the violin) there is visibly co-dependent interaction of both gross and fine motor control with dynamic interchange of roles depending on the musical context [45,51]. To complicate matters further, because fine arts performance (e.g., music) unfolds over time, during which all motor control must be guided, performers' motor behaviors must be part of a contextual, forward-planned process. At the same time, since it is undesirable for performers to merely act as mechanical executors of instructions on a printed page [67], they must maintain sufficient flexibility to adapt to and accommodate circumstantial developments. At the highest levels of complexity, adaptation will include immediate forward re-planning. Thus, music performance must be considered a process, not a reconstruction; a reality that creates significant, but not insurmountable, challenges for researchers who apply scientific techniques to fine arts performance. Fine arts performance places more demands on sensing technology than those activities that are dominated by gross motor control in sports. Therefore, wearable sensors are gaining endless interest nowadays in sports [5,7], whereas, they are still sporadic in arts performance [39,68].

The third challenge is the expert-knowledge (i.e., compensatory strategies depending on an individual anthropometry and physical condition) in complicated motor skills learning. Motor control in sports and arts are acknowledged to be activities requiring complex behavior and long-time motor control development [14,42,59]. Athletes and artists take significant amounts of training and practice for individualized development, i.e., motor skill optimization based on their body structures and physical uniqueness. During their years of training, the desirability of acquiring performance skills efficiently and effectively while simultaneously avoiding injury would seem self-evident. Therefore, athletes and artists at various levels are continually searching for opportunities to improve their motor skills and gain advantages or perfection in their competitions or performances. Study on developing individualized compensatory joint-coordination is still feeble.

In brief, developing real-time biomechanical feedback needs for searching ways/body models to supply information, which should consider the motor-control diversity, the anthropometrical variation, and physical compensation/optimization.

## 5. Significant Gaps in the Current Research

### 5.1. The Lack of a General Full-Body Biomechanical Model Supported by a Practical Number of Wearables

Currently, the most reliable and practical full-body model for quantifying complicated human motor-skills consists of 15 segments: head, upper trunk, lower trunk, upper arms, lower arms, hands, thighs, shanks, and feet (Figure 1). This biomechanical model requires ~40 inputs (i.e., 40 selected body points) for determining joints kinematics in order to reveal the motor control/limbs' coordination, and the model has been successfully applied in numerous 3D motion analyses of human virtuosity in sports and arts performance [69–75].

Notwithstanding the above successes, the development of real-time biomechanical feedback is still in his infant phase. A brief look at the following facts may reveal one of the gaps in the development. (1) the concept of applying a general full-body model (i.e., 3D skeleton) in human movement study can be traced back to the early seminal research of Johansson in 1973 [76], which demonstrated that a major joint positions can effectively analyse human behaviors. (2) The tremendous success of the full-body model in 3D quantification of various human movements have been widely seen in the past decades [69–75]. (3) The limited applications of real-time biomechanical feedback in sports are still using special partial-body models that are related to rowing (limbs' accelerations) [77], shooting (stability of center of gravity) [78], bicycling (the foot angle profile) [79], and swimming (temporal stroke phase identification) [80]. The above facts might indicate that, for developing a universally applicable device, the first missing piece might be a general full-body biomechanical model supported by wearables.

Obviously, it is wise to transfer the success of the general full-body model from 3D motion capture domain to wearables domain; nevertheless, it is nonrealistic and impractical to use so many wearables for rebuilding the full-body model. The question aroused is what is the minimum wearables required for the rebuilding? So far, there is a dearth in research on this issue. For reaching a general application of real-time biomechanical feedback in sports and arts performance, studies are needed to discover ways for reducing the number of the model inputs to a doable level in order to apply wearables in practice; and such a reduction should not trade off the current accuracy of 3D motion technology.

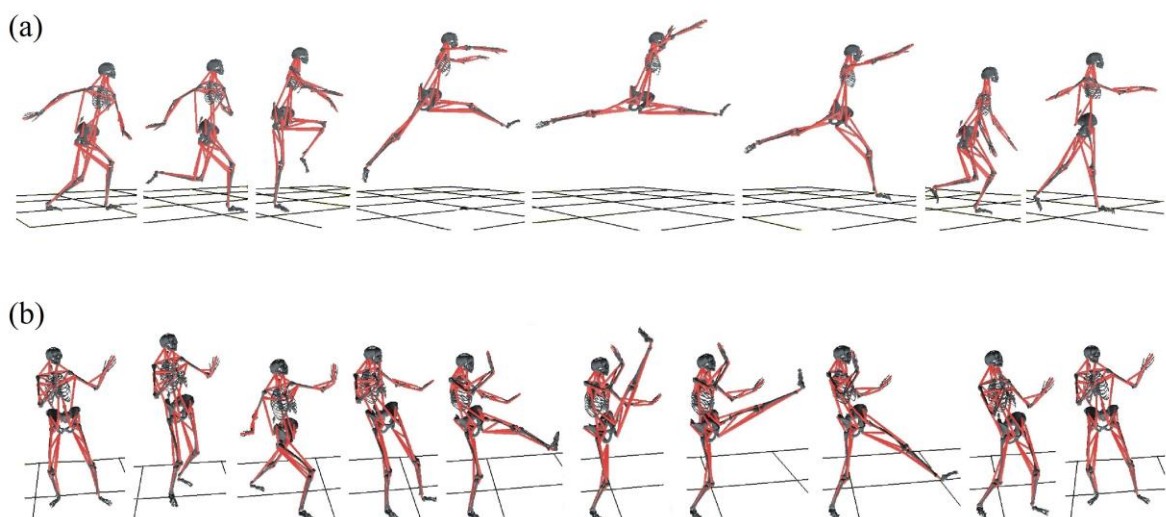

**Figure 1.** 15-segment biomechanical modeling of the Grande Jeté (**a**) in Ballet and the Axe Kick (**b**) in Taekwondo [56,81].

## 5.2. The Lack of a General Method for Identifying Motor-Control Patterns

It would be a practitioner's desire that, just like physiological, neurological, and biochemical wearables, the biomechanical one could also be universally applied to all motor skills for their learning and training in sports and arts. A general application means that a general method should exist for data interpretation (i.e., identification of motor-control patterns). Unfortunately, we are still far away from the goal. Apart the lack of a general full-body biomechanical model, there is not a general method for data interpretation. All existing studies are specific/isolate ones. So far, only few studies explored the real-time biomechanical feedback related to sport activities [64,82,83]. On one side, the feedbacks were force, power, or velocity; not the feedbacks of limbs' coordination, i.e., motor control patterns. On the other side, the evaluation methods were tailed to the activities being examined, i.e., no potential for delimitation.

Another confounding factor influencing the identification of motor-control patterns is anthropometry. In great part, the personal control 'style' derives from an individual development of a

variety of motor behaviors that are compensatory in nature; accommodating body size and shape as well as physical ability, such as flexibility and agility. This individualization, in turn, poses particular challenges for both research and pedagogy. An astonishing example of motor compensation can be observed in the performance of Canadian violinist Adrian Anantawan [84]. Adrian was born without a right hand, no wrist, and only a partial forearm. This means he has only two arm segments with two functional joints in his bowing arm—the arm responsible for much, if not most of the violinistic nuances that are associated with articulation and expression. A "normal" violinist would have the use of three large segments in the arm (upper arm, forearm, hand) plus the fingers (with a total of 17 functional joints) for the finest motor adjustments. Astoundingly, Adrian has found compensatory motor control mechanisms that allow him to accomplish performance at the highest of professional levels [85]. His accomplishments highlight that there are many motor control means to the same end. For meaningful motor control identification, novel methodology must be developed to examine its many facets simultaneously.

## 6. How Past and Current Developments Influencing New Endeavors—Ideas of Where Research Might Break Through

In area of human motor skill learning, people are always looking for ways to speed up training, ways to make it more economical, efficient, and effective, ways to minimize injuries. Real-time biomechanical feedback training could be the best way that people are looking for, because the technology would have potentials for: (1) making scientific monitoring from a lab-based environment to in field, (2) simplifying a scientific quantification from using a complicated motion capture system to easily-applied wearables, and (3) transferring the vital biomechanical feedback in a right time for preventing the worst/movement errors happening, while finding individual compensation/optimization [50]. For reaching this goal, further studies should focus on solving the challenges (Section 4.3) and bridging the gaps (Section 5). The research focuses could be summarized, as follows:

- Developing a wearable-based full-body biomechanical model that is equivalent to the current 3D 15-segment one
- Searching new method for wearable data interpretation, i.e., motor control characterization
- Generalizing the approach to various sports and arts performance.

The past and current studies of anthropometry, biomechanics, sensing technology, and AI have supplied rich food of thoughts that might be benefit to the future research endeavors.

### 6.1. A Two-Chain Model as a General Full-Body Biomechanical Model for Wearable-Data Collection?

Anthropometrical studies [38,86] show that an individual full-body model (equivalent to the generic 15-segment model in 3D motion capture) can alternatively be built indirectly using an anthropometrical approach. Using variables such as body weight, body height, age, gender, and race, one can statistically determine the segments' lengths, segments' masses, segments' COGs (center of gravity), and moment of inertia of body segments. Now, the question is how many wearables are needed to reliably determine the joints angles and limbs coordination? Additionally, the locations of wearables can introduce an artificial stimulus to the neurosensory system while measuring human movement, yielding motion patterns that do not reflect natural patterns of movement. Therefore, both the number and location of wearables are vital for developing real-time biomechanical feedback training.

Studies on these issues could be considered as no-theory-first type of studies. A common research strategy for developing a general method for a no-theory-first topic would be to begin with empirical inquiries on various cases (i.e., case studies) that investigate the subject within its real-life context [87]. The case studies could supply up-close, in-depth, and detailed aspects of the topic, as well as its related contextual conditions. Using inductive reasoning, research hypotheses could be made from these

case studies for future studies. As such, hypothesis-driven studies would be a logical, reasonable, and practical choice for making breakthroughs. It may be worth of trying this research path for the development of real-time biomechanical feedback.

Based on the current cases of 3D motion analysis studies [60,70,74,75,88,89], a variety of complicated human movements (both sports and arts performance) could be considered as a model system with two mechanical chains: upper-body chain and lower-body chain (Figure 2). The base segment of the upper-body chain is upper-trunk & head; two sub-chains (i.e., arms) are linked to the base. Similarly, the lower-body chain consists of a base (i.e., lower-trunk) and two sub-chains (i.e., legs). The proposed generic model of a human body would reduce the number of DOFs (degrees of freedom) in human movement quantification. With this abstraction, human motor-skills could be tracked by using fewer IMUs (inertial measurement units), a sensing technology that measures linear and angular motion usually with a triad of gyroscopes and triad of accelerometers [90–92]. Previous studies have proven the reliability and concurrent validity of the technology in human gait quantification related to clinical and/or physiotherapy applications and feedback training in rehabilitation [93,94]. It is time to explore the IMUs' potential in sports and arts applications [95].

For sports and arts applications, it would be possible, if one could apply three IMUs on each chain (one on the base, one on each distal end of the chain) (Figure 2a), to determine the segments'/joints' motion and coordination as well as the orientation–relationship between the two bases of the two chains. As such, human motor-skills can be quasi-naturally tracked (i.e., wearables minimally encumber human motor control), estimated, and recognized for the real-time biomechanical feedback training. Only, the IMUs' inputs are still not enough for using traditional ways (class mechanics and engineering) to quantitatively determine the model system. Yet, artificial intelligent (AI) could clear the barrier due to its "learning" ability [96,97].

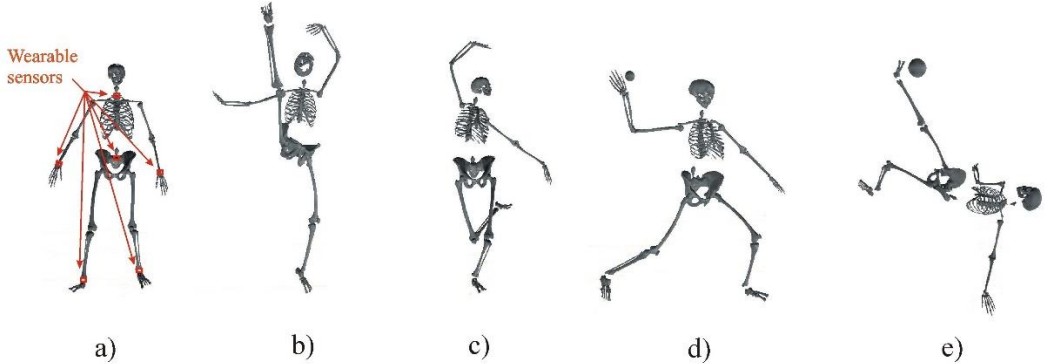

**Figure 2.** The two-chain model of human motor-skills. (**a**) The possible locations of the six wearables for human motor-skills' tracking; (**b**) A ballet skill; (**c**) A Indian dance skill; (**d**) Baseball pitch; and, (**e**) Bicycle kick in soccer (the three-dimensional (3D) motion data was generated in Shan's Biomechanics Lab).

### 6.2. AI for Motor-Control Quantification

The AI systems are performance driven—one focus is on the predictive accuracy, based on known characteristics learned from the previous data/training samples [98,99]. In the past decades, AI techniques have experienced a resurgence following concurrent advances in computer power, large amounts of data (big data), and theoretical understanding. AI techniques have become a powerful tool for helping to solve many challenging problems in human motor-skills' evaluations and analyses [96,97,100–102].

The idea of AI prediction is to find a way to learn general features in order to make sense of new data [98,99]. This description highlights the central role of data for establishing implicit knowledge. The amount of data must be sufficiently large to provide many training examples from which a large set of parameters can be extracted.

Among the AI technologies, deep learning is considered as a powerful tool that percolates through to all application areas of machine learning, such as image identification, speech recognition, natural language processing, and, indeed, biofeedback support [103–105]. The success of deep learning networks encourages their implementation in further applications for the enhancement of human physical activities [106,107]. Most recently (September 2018), Nature Neuroscience has published the latest developments in the area of markerless, video-based motion tracking, indicating that the power of deep learning will enable motion tracking to human-like accuracy [108]. This study confirms that motion capture/quantification of limbs' coordination will move from an expensive and difficult task restricted to the laboratory to an effortless daily routine for researchers and practitioners.

From motor learning point of view, wearables would have much higher potential than video shooting in the future practice. This is not only because of the fast advance in miniature of wearables, but also due to two inherited drawbacks of video-shooting approach. Reliable biomechanical feedback should obtain from accurate quantification of human movement in field, with some requiring large space. Even with a multiple-camera setting, unexpected environmental factors (e.g., interactions among athletes) will create data-gap. Further, it is true that we are already sitting on massive movement data (e.g., YouTube, Flickr) for training of deep learning models; but the video datasets are un-calibrated and have very little information on the hardware and conditions used to capture particular videos, which can bias the deep learning recognition algorithms [109]. Currently, the availability of reliable motion capture data for developing deep learning models is significantly limited.

Summarized from Sections 6.1 and 6.2, the combination of the two-chain full-body model with six wearable IMUs and the deep learning prediction based on IMUs' data shows great potential in developing real-time biomechanical feedback training for an efficient human motor-skill learning and optimization. The missing piece for testing the potential is reliable massive training data.

### 6.3. The Diversification of Deep-Learning Training Datasets for Increasing Feedback Reliability

The current studies show that two factors strongly influence deep learning performance [99,110–112]. One is the massive data and the other one is the diversity of the massive data. A systematical review article in 2018 has examined 53 studies (published from 1 January 2008 to 31 December 2017) on deep learning applications of the physiological data/signals in healthcare. The article found that, not only the amount of data, but also the diversity of data would affect the prediction's reliability [99]. This result would suggest that deep learning algorithms would perform well with large and diverse datasets.

It is well known that, among all human physical activities, sports and arts skills exhibit the most diversity of motor control. The datasets that are available for developing deep learning models have to reflect the diversity, because the depth and specialization must come from training the deep learning algorithms with the massive and diverse data collected from sports and arts motor skills. Therefore, at present, the vital step for developing real-time biomechanical feedback tool is to simultaneously collect a large amount of motion data using both 3D motion capture (e.g., the two-chain model with ~40 markers) and wearable IMUs (e.g., the same model with six IMUs). The datasets should cover large variety of sports skills and arts performances. As such, the 3D motion-capture data can be served as a "supervisor" for training network model to map IMUs data to joints' kinematic data. Such a deep learning model could be universally applied in motor learning and the training of sports and arts skills.

Retrospectively, the current knowledge (i.e., anthropometry, biomechanical modeling, and deep learning) and technology (i.e., miniaturized IMUs) supply an almost perfect environment for developing a real-time biomechanical feedback tool for general application in sports and arts. The missing piece is the massive and diverse motor-skill datasets for deep learning.

## 7. Conclusions

Through a review on the past and current state of biomechanical feedback studies in sports and arts performance, this paper introduced a two-chain model with six IMUs that are powered with

deep learning technology. The framework can serve as a basis for developing real-time biomechanical feedback training in practice. In order to creating a feasible, reliable, and practical biomechanical feedback tool for athletic and artistic motor-skills' learning and optimization, the massive and diverse motor-skill datasets have to be built first. The big data could be obtained by a synchronized measurement from 3D motion capture and IMUs. Currently, gaining high-quality, full-body motion data cross sports and arts performance would be the vital step for the real-time biomechanical feed-back development.

**Author Contributions:** X.Z., G.S., Y.W., B.W. and H.L. performed the literature search and design of the article; X.Z., G.S. and Y.W. prepared the draft; X.Z., G.S., Y.W., B.W. and H.L. contributed to the revisions and proof reading of the article.

**Funding:** This research was funded by National Sciences and Engineering Research Council of Canada (NSERC), grant number RGPIN-2014-03648 and Xinzhou Teachers' University/China.

**Conflicts of Interest:** The authors declare no conflict of interest. The founding sponsors had no role in the collection, analyses, or interpretation of data; in the writing of the manuscript, and in the decision to publish the results.

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
