# Peer review of "Wearables, Biomechanical Feedback, and Human Motor-Skills’ Learning & Optimization"

_applsci, doi:10.3390/app9020226_

Reviewer 1 Report

Wearable, Biomechanical Feedback and Human Motor-skill’s Learning & Optimization

Authors provide a comprehensive review of biomechanical feedback in the field for improving sports and arts performance has been reviewed. Authors pointed a real-time biomechanical feedback application has been limited, so four specific areas including 3-D motion capture, anthropometry, biomechanical modeling, sensing technology, and decision-making technique. Based on their review, the authors proposed a two-chain body model integrating IMUs and deep learning-based analytics. The six milestones between chapter 1 through 5  in the existing research to understand current technology status were well organized to provide a solid conclusion.

However, the authors’ argument regarding the gaps in the current research seems quite optimistic.

First, the lack of a general full-body biomechanical model is not supported by evidence. Beside authors suggestion, there is no objective rationale that the two-chain model with six IMUs is beneficial. I expect the authors any comparison or preliminary experimental results that the model in the manuscript is practical for any purpose in the sports and arts performance field.

Second, a term in the title (i.e., Deep learning) seems irrelevant to the whole manuscript. I expected a specific application example that illustrate any preliminary level application and outcomes from the application. The current manuscript is not showing any feasibility that the two-chain model and deep learning could address the limitation authors pointed.  I would also expect authors could provide why the deep learning-based analytics is beneficial than rule-based learning approaches.

Best regards, 

Author Response

Reviewer’s comment:

Authors provide a comprehensive review of biomechanical feedback in the field for improving sports and arts performance has been reviewed. Authors pointed a real-time biomechanical feedback application has been limited, so four specific areas including 3-D motion capture, anthropometry, biomechanical modeling, sensing technology, and decision-making technique. Based on their review, the authors proposed a two-chain body model integrating IMUs and deep learning-based analytics. The six milestones between chapter 1 through 5  in the existing research to understand current technology status were well organized to provide a solid conclusion.

RESPONSE:

We appreciate the reviewer’s overall evaluation very much.

Reviewer’s comment:

However, the authors’ argument regarding the gaps in the current research seems quite optimistic.

First, the lack of a general full-body biomechanical model is not supported by evidence. Beside authors suggestion, there is no objective rationale that the two-chain model with six IMUs is beneficial. I expect the authors any comparison or preliminary experimental results that the model in the manuscript is practical for any purpose in the sports and arts performance field.

RESPONSE:

Thank you. The suggestion is well taken. The following revisions have been made.

Regarding the lack of a general full-body biomechanical model, a paragraph with 5 new references is added to supply the supporting evidences (please see p 6-7).

As for the rationale, we have rewritten the session 6.1. Basically, for developing breakthrough studies in the future, this review article aims to make hypothesis (rather than rationale) for initiating novel studies. We have added more elaborations on this point and softed our tone (from “can” to “could”) for avoiding misunderstanding (please see p 8-9).       

Reviewer’s comment:

Second, a term in the title (i.e., Deep learning) seems irrelevant to the whole manuscript. I expected a specific application example that illustrate any preliminary level application and outcomes from the application. The current manuscript is not showing any feasibility that the two-chain model and deep learning could address the limitation authors pointed.  I would also expect authors could provide why the deep learning-based analytics is beneficial than rule-based learning approaches.

RESPONSE:

Thank you. The concern is related to the previous one. Research in this area could be considered as a no-theory-first type of study. Researchers are still searching for potential breakthroughs. Logical and scientifically-reasonable hypotheses would be desired for launching novel studies, which would lead to the “preliminary level applications”. We believe that our rewritten (p 8-9) has addressed this concern.

Reviewer 2 Report

Dear Authors,

Thank you for this revised draft. It has been improved.

Reliability and validity of  wearable systems are important aspects that should be discussed fully highlighted and discussed. This is because these aspects are important when we need to measure movement analysis and evaluate sport performance. I recommend the authors to include the recent literature about this topic. Examples include:

Inertial Measurement Units for Clinical Movement Analysis: Reliability and Concurrent Validity

Xsens MVN: Consistent Tracking of Human Motion Using Inertial Sensing

Validation of wearable visual feedback for retraining foot progression angle using inertial sensors and an augmented reality headset

Author Response

Reviewer’s comment:

Thank you for this revised draft. It has been improved.

Reliability and validity of  wearable systems are important aspects that should be discussed fully highlighted and discussed. This is because these aspects are important when we need to measure movement analysis and evaluate sport performance. I recommend the authors to include the recent literature about this topic. Examples include:

Inertial Measurement Units for Clinical Movement Analysis: Reliability and Concurrent Validity

Xsens MVN: Consistent Tracking of Human Motion Using Inertial Sensing

Validation of wearable visual feedback for retraining foot progression angle using inertial sensors and an augmented reality headset.

RESPONSE:

Thank you. The suggestion is well taken. Revisions have been made for discussing the reliability and validity of IMU and the suggested article are added for supporting the revised discussion (please see p9).

Round  2

Reviewer 1 Report

The authors have added more elaboration to improve the quality of the manuscript for avoiding misunderstanding. I nominate the revised manuscript to publish in a present form.